# Learning Objects in the Educational Context: The Perspective of Teachers in the Azores



Ana Isabel Santos [1,*], Ana C. Costa [2,3], Andrea Z. Botelho [2,3], Manuela I. Parente [2,3], José Cascalho [3,4], Diana Freitas [5], André Behr [5], Ana Rodrigues [2] and Armando B. Mendes [3,4]

1   Faculty of Social Sciences and Humanities, Interdisciplinary Centre for Childhood and Adolescence, University of the Azores, 9500-321 Ponta Delgada, Portugal
2   CIBIO—Research Center in Biodiversity and Genetic Resources/InBIO—Associate Laboratory, University of the Azores, 9500-321 Ponta Delgada, Portugal; ana.cm.costa@uac.pt (A.C.C.); andrea.zc.botelho@uac.pt (A.Z.B.); manuela.ip.cardoso@uac.pt (M.I.P.); ana.mf.rodrigues@uac.pt (A.R.)
3   Faculty of Sciences and Technology, University of the Azores, 9500-321 Ponta Delgada, Portugal; jose.mv.cascalho@uac.pt (J.C.); armando.b.mendes@uac.pt (A.B.M.)
4   GRIA and Artificial Intelligence and Computer Science Laboratory, 4150-181 Porto, Portugal
5   NIDeS and University of the Azores, 9500-321 Ponta Delgada, Portugal; diana.ca.freitas@uac.pt (D.F.); andre.r.behr@uac.pt (A.B.)
*   Correspondence: ana.is.santos@uac.pt

**Abstract:** This paper seeks to identify the pedagogical resources used by kindergarten, primary and secondary teachers in Azores Islands. Additionally, an investigation will be made into how these resources are mobilized in teachers' pedagogical practice, with the aim of understanding to what extent digital resources, particularly learning objects, are present in schools. For this purpose, a study was developed, which included a questionnaire survey conducted online, and sent to teachers in 2021/22. A total of 426 answers allowed us to conclude that the use of pedagogical resources is still far from the current and emerging need to mobilize digital resources, particularly learning objects, as a tool to enhance meaningful learning.

**Keywords:** pedagogical resources; learning objects; digital educational resources; teacher's perspectives

## 1. Introduction

For decades, teachers from all educational levels have been using resources to prepare and implement their pedagogical work. This has been done in a way that the teachers consider appropriate and adequate to their social and cultural reality, regulations, guidelines, and curricular demands imposed upon them. Often, these resources are very focused on the teacher and teaching processes rather than on learning. The teaching resources used are, in most cases, physical, paper-based objects, such as textbooks [1].

However, the growing digitalization of society and consequently the contexts in which children and young people live, have lead to a learning process that is no longer exclusively linked to the teacher's action in the classroom in the familiar logic of teacher vs. learner.

From a student-centered learning perspective, the inclusion of digital resources in pedagogical practices seems to be an enriching pedagogical method because it enables access to diverse resources and content. In particular, the use of learning object (LO) repositories by teachers and students seems to be a key learning tool nowadays. This is considered to be appealing and motivating for students, and at the same time, interesting for teachers because of the potential for specific content searching that it guarantees [2,3].

In a remote archipelago with a considerable distance among the islands of the Azores, digital resources can be a good solution for teachers as the access to other teaching solutions is not as easy as in mainland urban centers. However, there is an empirical perception that these methodologies are not fully exploited by secondary school teachers.

This study seeks to carry out a diagnostic assessment of which and how pedagogical resources are used by teachers in the Azores, especially to understand the motivations and effective use of digital-based resources vs. non-digital based resources. It also intends to identify and analyze the type of pedagogical resources used in the Azores by teachers from all school levels, from kindergarten to secondary education. We pretend to know how they mobilize these in their pedagogical practice, seeking to understand to what extent digital resources, particularly learning objects, are present in Azorean schools. This is particularly relevant to assess the usefulness and relevance of the development of a web-based repository of digital learning objects to increase Ocean Literacy, available in re-mar.uac.pt (accessed on 31 March 2022) [4,5], that was developed under the seaThings project. As far as we know, there have been no studies in the Azores on the use of LOs. This study may represent an instrument for political decision, concerning teacher training in a region where educational policies have dependent within local government.

In the following section, we review the literature on the subject of the use of digital pedagogic resources and learning objects. Next, we present the methodological approach followed by the main results of the survey. The text ends with a discussion and conclusions.

## 2. Literature Review

### 2.1. From Pedagogical Resources to Digital Literacy

The educational system in the 19th century, according to [6], defined learning as a passive activity where the teacher transmitted the knowledge and the students, sitting in silence, received the information and made their notes. However, already in that century, several pedagogues attributed some relevance to teaching materials for children's development and learning. There was a call for active, constructive, and meaningful learning, made by educators such as Frederick Froebel or Maria Montessori, who are still relevant today [7].

In a teacher-centered educational processes, the student receives information and knowledge passively by "sitting, listening, copying, memorizing and repeating what the teacher has said" without the need for teaching resources other than paper and a pencil for the student and a board for the teacher. In an active, student-centered intervention, teachers and students work together "to select teaching objectives and tasks based on real problems", considering what students already know, their previous experiences, and interests. Students are confronted with the contextualized use of competencies that allow "learning to know, to do, to speak and to know sciences … " [8] (p. 74), using a range of resources to meet those formative needs and interests.

This active, experiential, manipulative use of pedagogical resources, teaching materials, or didactic resources, as they may be called, brings about the logic of decentering the learning processes from the teacher to the student. As [9] points out, it is important that the student is interested and involved in learning, stimulating research and the search for new knowledge. Digital technologies have invaded the classroom in the last decade, as students have incorporated the Internet into learning, whether as a part of a pedagogical proposal or not [10]. Nowadays, formal education relies on the integration of educational technology, both to deliver content (e.g., [11]) and as a result of teaching and learning efforts (e.g., [12,13]). Therefore, concepts such as digital competence, digital literacy, or digital skills are of growing importance [14]. Moreover, [15] demonstrated the learning effectiveness of digital tools against non-digital ones in the context of museum exhibitions.

Unfortunately, textbook-centered practices continue to be referenced as very common in studies such as that of [1], even though teachers highly value materials that are self-made because they consider them to enhance meaningful learning, and because, admittedly, they make a contribution to their professional development.

On the other hand, as [16] points out, in the current digitalization of society, the idea that only qualified personnel can teach is increasingly being put aside. The use of technologies and digital reinforces the idea that everyone, regardless of age or qualification, can teach, because knowledge is available on the web and learning can occur in different contexts.

Nevertheless, this situation does not discharge the relevant role of the teacher as a "learning designer or facilitator" for designing and implementing learning situations that allow students to make better use of the digital resources available, seeking to enhance their digital literacy.

The COVID-19 pandemic situation brought a greater need to develop skills that foster digital literacy in children and young people [11]. Ref. [17] (p. 59) reveal that "during their study at home, they spend much time surfing the internet, either through cell phones, personal computers, or laptops", but "... there are still many students who are not wise in using the internet such as spreading hoaxes, citing articles without the author's permission and many others". Developing the skills to choose more carefully and discerningly the sources of information to consult, to be more proficient in using technological resources, analyzing correctly mobilizing the information provided by digital resources are key competencies that embody the concept of digital literacy [17,18].

For these transformations to take place in school, teachers must envision themselves as "curriculum developers", as [19] (p. 42) states, which implies:

- "a professional vision on and responsibility for the curriculum which inform the educator's thinking about pedagogic principles;
- the creation of consistency and coherence within the curriculum;
- curriculum innovation through the application of the latest theoretical and practical insights and developments;
- materials development".

These principles can, combined with a logic of 'collegial collaboration' between teachers, as [19] point out, be valuable contributions to the construction of professionalism that better response to current demands. When this happens, and when resources are mobilized effectively, teachers do not only demonstrate their understanding of the curriculum but also respond better to students' characteristics and needs [20].

### 2.2. Learning Objects as Tools at the Service of Teachers and Students

In the perspective of promoting digital literacy, learning objects (LOs) emerge. There are several definitions for LOs, although the most widely used has been proposed by the Institute of Electrical and Electronics Engineers/Learning Technology Standards Committee (IEEE LTSC), which defines LOs as any digital or non-digital entity that can be used and reused and is available on the web (IEEE LTSC, 2010). Other authors emphasize characteristics as reusable and self-sufficient tools, described by metadata, which allow aggregation within other collections, carrying learning objectives in themselves, as they are focused on specific subject areas and concepts of the curriculum, and which are small and easy to use [3,21–23]. Examples of LOs can be electronic texts, multimedia content, video games, educational games, websites, or images.

Due to the richness of the information contained in LOs, they can be a relevant resource for education by facilitating the opening of new horizons for the pedagogical work of teachers and students, allowing the improvement of learning processes [24]. LOs' collection in a repository brings the advantage of their usage by teachers, students, or other educational agents according to their needs or interests, since their organization in terms of, for example, file format, subject, grade level, or language, allows it. At the same time, their use facilitates the sharing of LOs created by teachers and students themselves, which seems to be a positive aspect in teachers' opinion, as they can also receive feedback from colleagues [2].

Simultaneously, LOs used in educational settings leads to a paradigm shift, as they are considered "an excellent tool in inverted settings of instruction as they enabled a shift in the classroom dynamics towards a learner-centered approach" [25] (p. 2961). As also pointed out by [3] (2021), when designed in this constructivist, learner-centered perspective, LOs are more likely to motivate and interest students and, at the same time, they are an important tool for teachers, by enhancing interactivity and engagement.

Thus, using LOs presupposes access to activities that promote the dynamic and autonomous acquisition of knowledge, breaking more traditional teaching approaches, centered on the teacher [26], seeking, with this use, to facilitate and improve the quality of the learning processes that take place in an educational context.

Despite the wide availability of educational resources, the task of selecting relevant educational resources is arduous and exhausting for teachers and students [27]. The Internet is full of educational resources, which can become an obstacle to learning processes since teachers and students spend more time looking for content to study rather than studying. Usually, LOs are catalogued in repositories that help to improve their use and reuse [24]. However, most teachers using digital Internet resources are not aware of learning objects, object repositories, or how learning objects are developed. For the right use of these learning resources, some concern is needed in the development and identification of LOs, as well as a focus on teacher training [28].

## 3. Materials and Methods

### 3.1. Objectives

The main objective of this study is to assess how teachers use pedagogical resources in the Azores, where the distance between islands and national main urban centers has long impacted populations and consequently teaching.

The digitalization of teaching can be particularly relevant to overcoming issues posed by physical distances. It is then important to understand the motivations and effective use of digital-based resources vs. non-digital-based resources by the local teaching community. Furthermore, the authors aim to evaluate the receptiveness of a web-based repository of digital learning objects to increase Ocean Literacy, available in re-mar.uac.pt (accessed on 31 March 2022) [4,5] that was developed under the Seathings project, to seek strategies to encourage its use and therefore promote ocean literacy.

In this sense, the specific objectives of the study are:

- To analyze how teachers mobilize pedagogical resources in their teaching practice;
- To identify the type of resources, digital or non-digital, that teachers use most frequently in their teaching practice;
- To identify the reasons that lead teachers to use certain types of resources to the detriment of others;
- To analyze possible differences in the use of pedagogical resources among teachers, considering different educational levels (early childhood, primary, high school and secondary education), teaching experience, subject areas and demographic variables (e.g., age, gender).

The present study's hypothesis is that science teachers from higher teaching levels (secondary) would be more prone to use digital resources than those teaching younger students and non-science subjects.

### 3.2. Instruments and Procedure

To survey the type of resources teachers use in their daily educational lives and how they use them, a questionnaire survey with open- and closed-ended questions was designed and applied defined as an effective instrument for describing a population [29].

According to information made available by the Regional Secretariat of Education, in 2019–2020 a total of 5502 teachers were in the Azores educational system, 73.2% female and 26.8% males, with a higher percentage in the 3rd cycle of basic education (CBE) and secondary education (39.5%), followed by 2nd cycle of basic education (19.5%) and 1st cycle of basic education (18.0%).

Based on this number of teachers, a minimum sampling number was calculated of 360 questionnaires for a binomial test with 95% confidence level.

The questionnaire is composed of three sections: (a) the first one characterizing the participants; (b) the second focus on the type of pedagogical resources and learning objects that teachers report using more often for the design and implementation of pedagogical



work, seeking to explore the purpose and the way they are used; (c) the third one focusing ocean literacy, which will not be the object of analysis in this paper. The questionnaire is available for consultation upon request to the first author.

The data was collected online, through Microsoft Forms, and sent to all teachers in the Autonomous Region of the Azores through the Regional Secretariat of Education, which oversees the Region's public schools. In addition to the ethical issues being safeguarded through the opinion of the Ethics Committee of the University of the Azores, they were also safeguarded through this form of communication.

We used the inductive content analysis technique to analyze open-ended answer, i.e., with categories that emerged from the answers obtained [30].

A one-way analysis of variance (ANOVA) was used to test the research hypothesis of whether there are any statistically significant differences in the frequency of use of digital or non-digital resources (dependent variable) as a function of age, teaching levels, geographical location (islands groups) and discipline/subject area (independent variables), using the IBM SPSS software version 27.

## 4. Results

### 4.1. Participants

A total of 426 kindergarten educators and teachers of basic and secondary education in the Autonomous Region of the Azores took part in the study (representative of 7.7% of the Azores teachers). Sixty were kindergarten teachers, 88 1st cycle of basic education teachers, 52, 2nd cycle of basic education, 131, 3rd cycle of basic education and 95 teachers of secondary education, all working in public schools in the Region. By subject areas, teachers in the 2nd and 3rd cycles of basic and secondary education are distributed according to Table 1.

**Table 1.** Distribution of 2nd, 3rd CBE and secondary school teachers by subject area.

| Subject Area | 2nd CBE | 3rd CBE | Secondary |
|---|---|---|---|
| Social Sciences and Humanities (Portuguese, History, Geography, English, German, French, Classical Languages, Economy and Accounting, Philosophy, Psychology, Moral and Religious Education) | 26 | 59 | 47 |
| Exact Sciences (Mathematics, Nature Sciences, Biology, Geology, Physics and Chemistry, Agriculture and Livestock) | 12 | 53 | 31 |
| ICT (ICT, Informatics, Technological Education, Electrical) | 2 | 6 | 3 |
| Special Education | 1 | 3 | 1 |
| Physical Education | 5 | 4 | 10 |
| Health | | | 1 |
| Artistic Expressions (Musical Education, Visual Education, Visual Arts) | 6 | 6 | 2 |
| total | 52 | 131 | 85 |

Fifty-eight of the respondents were working in the Eastern islands' group of the Archipelago, 41% in the Central group, and 2% in the Western group. The average age is 47.51 years and the average number of years of teaching service is 21.52 years. Most of them have a bachelor's degree (Figure 1).

Of all respondents, 46% (198) stated that they had attended some training or course related to information and communication technology in the last two years.

### 4.2. The Perspective of Teachers in the Azores about the Used Pedagogical Resources

The teachers in the region were asked about the type of resources they use when planning their pedagogical work. The most identified resources were the Curricular Guidelines and Official Programs, which dictate what should be covered at each grade level and school year, closely followed by textbooks. We found the videos and platforms that offer specific content to be less popular (Figure 2).

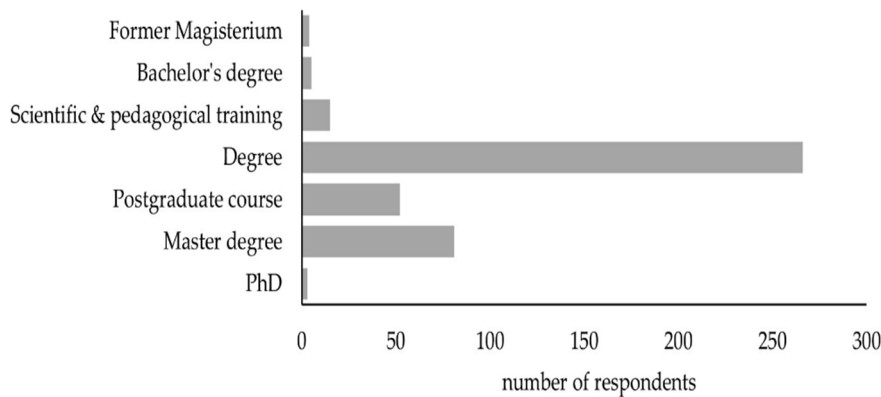

**Figure 1.** Respondents' Academic Qualifications.

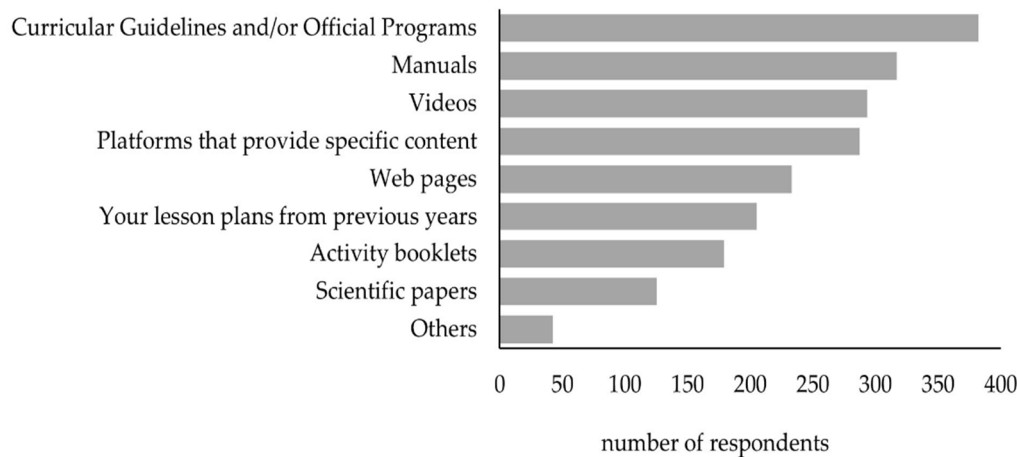

**Figure 2.** Resources used in the planning of pedagogical work.

Of the 42 teachers who reported using other resources, the ones they most commonly use belong to two major categories. The first focused on non-digital resources, associated with a more traditional educational perspective, where paper resources predominate, such as textbooks, books, newspapers, magazines, materials constructed by teachers that respect the students' interests, and audio and video resources. The second category, less often mentioned by teachers, points to digital resources and tools to support planning. The information contained in Table 2 is clear, showing a strong reference to various non-digital resources.

**Table 2.** Other resources used in planning.

| Non-Digital | | Digital | |
|---|---|---|---|
| Children's interest (questionnaire, students' questions) | 6 | Various applications (Kahoot, Educa Play and Learning Apps, etc.) | 6 |
| Teacher-made materials | 5 | Digital presentations (PowerPoint) | 5 |
| Books | 4 | Online information | 3 |
| Specific/scientific journals | 4 | Specific software | 1 |
| Movies/Documentaries | 4 | | |
| Audio Resources | 3 | | |
| Newspapers | 2 | | |
| Teaching resources from textbook publishers | 2 | | |

**Table 2.** *Cont.*

| Non-Digital | | Digital |
| --- | --- | --- |
| Tests/examinations | 2 | |
| TV programs | 2 | |
| Other colleagues' materials | 2 | |
| Photographs/Images | 2 | |
| Resources for training courses | 1 | |
| Training plans | 1 | |
| Manipulable material | 1 | |
| Teacher summaries | 1 | |
| External experts | 1 | |
| Students' productions | 1 | |
| Local institutions | 1 | |

Resources are chosen mainly because they offer relevant and valid information for planning (44%) and meet the current syllabus guidelines (35%), as illustrated understand why teachers preferred certain types of resources over others (Figure 3).

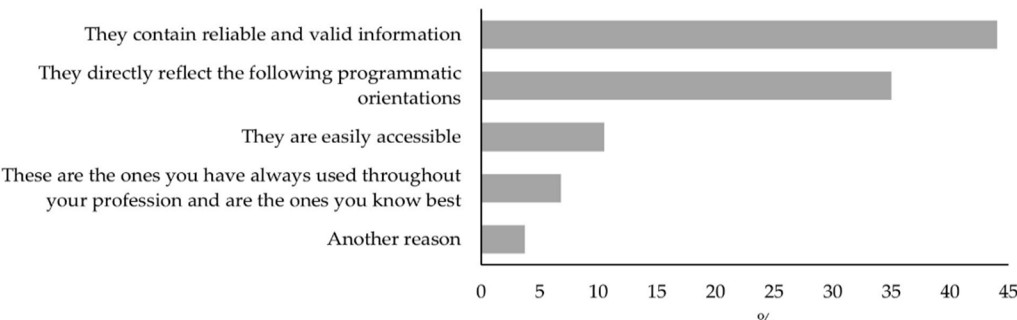

**Figure 3.** Reasons for the use of resources in the planning of pedagogical work.

Other reasons given relate to the need to promote students' interest (5 answers), to adjust to the demands and requests of the profession and students (5 answers), to fit in with the methodologies followed by teachers (3 answers), to diversify intervention strategies (1 answer) and to enable students' exploration or consultation (1 answer). In general, strategies focus on the teacher and the work he/she does.

Moving from planning to action, we tried to understand which resources they use most frequently to implement their teaching practice, i.e., in their day-to-day work with children and young people. As shown in Figure 4, textbooks or manuals are the resources used in most classes by teachers for the work to be developed. Activity sheets (70.7%) and videos (70.2%) were frequently used, associated with more expository intervention strategies. However, digital platforms that provide specific content (62%) and web pages (60.9%) are also frequently used. Films, games, and books other than textbooks are used infrequently during the school year.

Some respondents indicated other resources they consider frequently used in their pedagogical action (Table 3) either non-digital as field trips or own produced materials or digital, e.g., presentations and mobile apps but the predominant use of non-digital resources in the respondents' pedagogical action is also clear among these results.

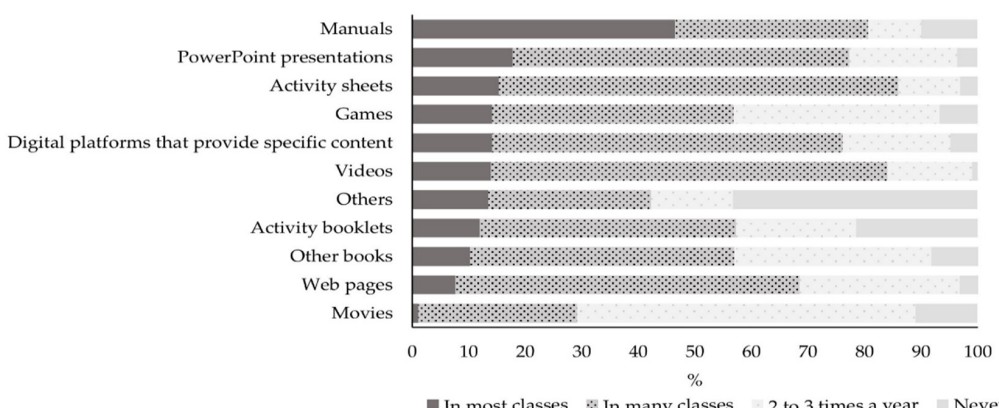

**Figure 4.** Resources used in pedagogical practice.

**Table 3.** Other resources used in pedagogical practice.

| Non-Digital | | Digital | |
|---|---|---|---|
| Study visits/field trips | 9 | Digital presentations (PowerPoint) | 3 |
| Teacher-made materials | 8 | Various applications (Kahoot, Educa Play, Learning Apps, etc.) | 3 |
| Manipulable material | 5 | Interactive whiteboard | 1 |
| Students' productions | 4 | Digital resources created by teachers | 1 |
| Forms created by teachers | 3 | Online information | 1 |
| Research/Projects | 3 | | |
| Practical assignments | 3 | | |
| Newspapers | 2 | | |
| Books | 2 | | |
| Films/Documentaries/TV shows | 2 | | |
| Tests/examinations | 1 | | |
| Experts | 1 | | |
| Games | 1 | | |
| Models | 1 | | |
| Resources for training courses | 1 | | |
| Board | 1 | | |
| Specific/scientific journals | 1 | | |

The reasons for using these resources are mostly related to the fact that they provide relevant pedagogical information, which allows teachers to transpose it into their practice without major changes and contain reliable and valid scientific information (Figure 5). Less important is the fact that they are constructed by the teacher him/herself, which may be more suitable for a group of students and are accessible for the students to use. The reasons "they were the ones that the teacher had always used during his/her profession" and "they are easily accessible" were not very important.

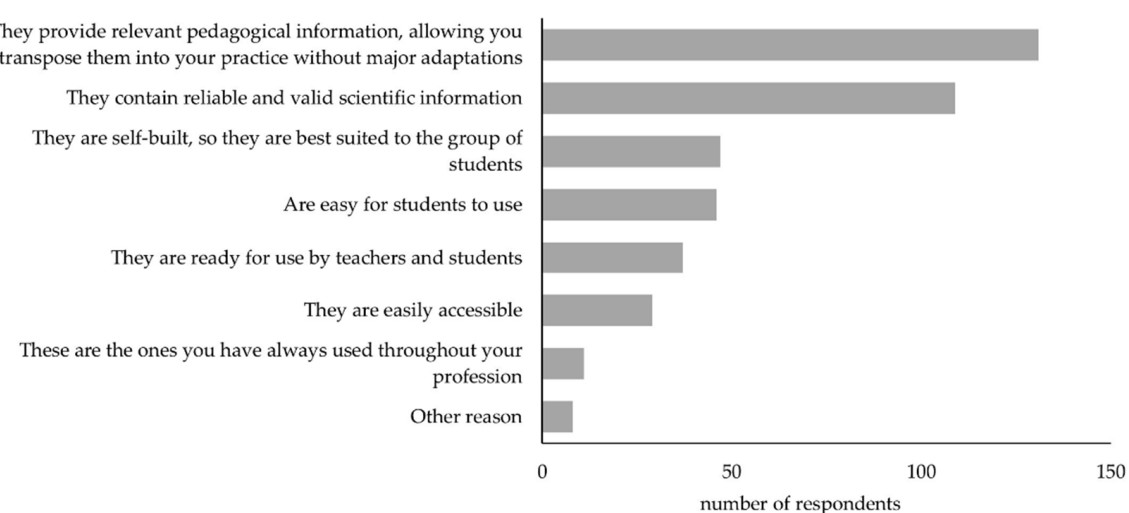

**Figure 5.** Reasons for the use of resources in pedagogical practice.

Other reasons stated for using teaching resources in practice are that they can be more motivating for students (1 answer), more appropriate to the pedagogical methodology followed by the teacher (1 answer), because they allow more flexibility (1 answer) and for all the above reasons presented in Figure 5 (5).

Even though it could be expected that younger teachers would use more digital platforms and resources, any significant difference was not found in digital resources usage according to age. Age did not contribute to differences in non-digital resources usage, and the manuals were still the most used items. Also, no differences were found in the use of digital and non-digital resources regarding teachers' teaching levels, subject area, or islands groups in the use of digital and non-digital resources (Table 4).

**Table 4.** Univariate analysis of variance (ANOVA) of non-digital and digital resources.

|  | **Non-Digital** | | | | **Digital** | | | |
| --- | --- | --- | --- | --- | --- | --- | --- | --- |
|  | df | Mean Square | F | Sig. | df | Mean Square | F | Sig. |
| Age | 18 | 1.851 | 1.204 | 0.254 | 19 | 1.718 | 1.114 | 0.333 |
| Curriculum area | 18 | 3.405 | 1.326 | 0.167 | 19 | 1.996 | 0.758 | 0.757 |
| Teaching level | 18 | 2.870 | 1.543 | 0.072 | 19 | 1.610 | 0.840 | 0.659 |
| Island Group | 18 | 0.326 | 1.171 | 0.282 | 19 | 0.097 | 0.336 | 0.997 |

Teachers' resources are mobilized, in most classes, for the teaching contents (51.6%) and their research as teachers (32.4%). In many lessons, the used resources seek to provide support materials for students (64.4%) serve for student research (59.6%), for students to carry out assignments (54.2%), and as a tool to assess students' learning (53.9%). With little expression throughout the school year, we identified the use of resources to disseminate the work done by students, which collected 44% of responses in the category "2 or 3 times a year" (Figure 6).

In addition to the resources listed in the questionnaire, respondents indicated other teaching resources that they considered could contribute positively to developing the teaching component of their work. The data presented in Table 5, while continuing to reinforce the need for non-digital resources, some of them aimed at the teachers' needs, reveal a growing need to equip and improve the conditions offered at schools, such as the need for computers, projectors, and Internet access under the best conditions. Among the digital resources listed, we highlight the need to have access to various resources—"appealing digital resources", "digital materials from publishers", and "digital educational games in European Portuguese (not Brazilian Portuguese)". Nevertheless, the need to

know and have access to databases and digital platforms that offer organized information and with search standards that can lead teachers to find specific information, as shown in the following answers: "digital content directed to the exploration and development of projects in kindergartens", "a digital repository with access to specific content", "paid digital platforms that provide content and educational resources".

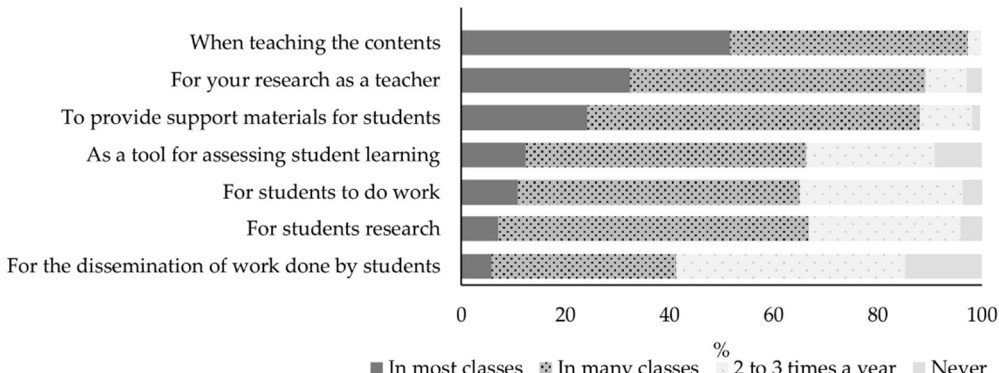

**Figure 6.** Use of resources in pedagogical practice.

**Table 5.** Resources that could contribute to enhancing pedagogical practice.

| Non-Digital | | Digital | |
|---|---|---|---|
| Play material/games/manipulative material | 20 | Digital resources | 15 |
| Study visits/field trips | 13 | Online databases/Platforms organized by content | 10 |
| Liaison with external entities (e.g., university professors, specialists, community) | 9 | Good quality internet in the classrooms | 6 |
| Audio and video materials | 8 | Smartboards | 6 |
| Books | 5 | Laptop computers/computer equipment in the classroom | 6 |
| Spaces equipped for specific purposes | 3 | Video projector | 5 |
| Lectures/dissemination sessions | 2 | Specific software | 3 |
| Real models | 1 | Smart games | 2 |
| Specific training courses | 1 | | |
| Teacher's guide | 1 | | |
| Protocols for laboratory/experimental/practical activities | 1 | | |
| Bilingual resources (Portuguese Sign Language) | 1 | | |

The teachers in the Region would be attentive and available to use, in their work in the classroom context, a digital repository that would allow them to have access to learning objects directed to specific contents and that, at the same time, would allow them to disseminate some of their work and the work of their students, as 350 (85%) suggested that they would be available to use it. Only 62 (15%) pointed out that they would not.

For most of those willing to use this type of resource in teaching practice, it would make sense to use it jointly by teachers and students (73%). With similar percentages and with less expression, the following uses were considered: for use by students (7%), for

use as teachers (7%), as a platform to disseminate information (6%) and as a platform to download information (7%) (Figure 7).

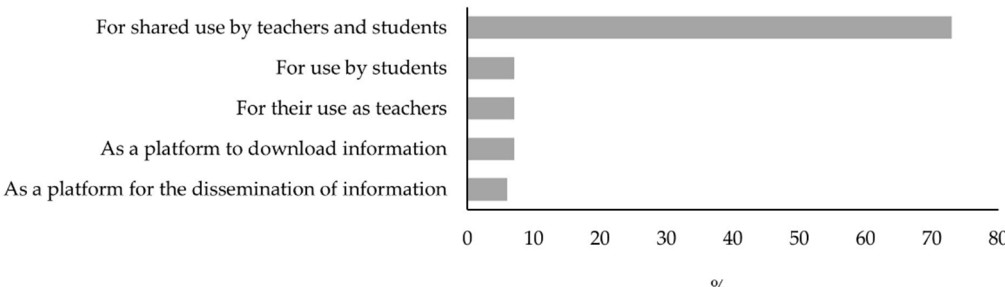

**Figure 7.** Use of a digital repository.

Amongst the reasons given by teachers for not being willing to use digital repositories, the time-consuming nature of such resources within the time available for lessons (30 responses) was the most mentioned. In total, 13 of the teachers indicated that they did not master digital technologies and 8 indicated that there were no conditions at school for web access.

In addition to these justifications, others were listed and related to licensing use and the care to not commit plagiarism (2 answers), the low adherence and lack of interest on the part of students for their use (2 answers), as expressed by one of the teachers that " . . . digital is not as appealing to the student as they would like us to believe lately [and that] the teacher needs to dedicate their class time to the student whether with a whiteboard, a computer or in a notebook", personal and professional reasons (4 answers) and the nature of the subjects they teach (2 answers). In this last group of answers, one was particularly interesting as it seemed contradictory to the current pedagogical relevance that the use of technologies assumes in children's development and learning processes with special educational needs. The teacher states that "they are students in the Special Educational Scheme and have high technological limitations".

Finally, we sought to understand if teachers had access to a digital platform that offered them useful information for their professional activity, and what would they like to find (Figure 8). Most of the respondents were looking for a platform that allows them to access diversified teaching resources (157 answers), which offers scientific and pedagogical information (108 answers), and which provides access to diversified teaching intervention strategies (71 answers). Less considered were the following factors: allowing a quick and effective search considering different parameters (e.g., theme, school year, curricular area, keyword), with 57 answers; offering scientific information, with 11 answers; offering pedagogical information, with 4 answers; and allowing access to useful links, with 3 answers.

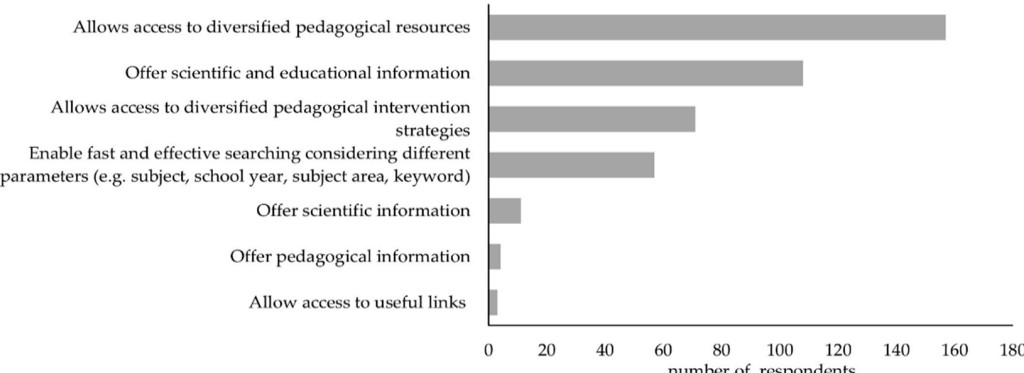

**Figure 8.** What teachers look for in a digital repository.

## 5. Discussion and Conclusions

Teachers in the Azores have incorporated digital resources into their teaching materials and no significant differences were found between digital and non-digital resources usage by the interviewed teachers related to age, experience, teaching levels, subject taught, or geographical location. Similarly, [31] found that university teachers in Latin America use digital tools regardless of the area of knowledge and did not find significant differences between the self-perceptions of males and females regarding the use of digital tools. However, in their study, gender has been shown to be a discriminating variable when crossed with the teaching experience, as females with more than 15 years of teaching experience undervalued motivation to use digital resources with respect to male teachers with the same teaching experience. In Ecuador, the level of digital competences in university teachers have been shown to be gender-independent but generation-dependent [32].

The recent effort made to increase technologies' usage during the pandemic [31] and the increase of digital educational resources available might have contributed to the more general use of these resources.

Also, it is important to notice that the sample was not entirely random. In fact, the procedure described corresponds to the most common form of nonprobabilistic sampling, normally known as convenience sampling, since no effort was made to fulfill control quotas. As the survey was distributed by an email link to an on-line form, it is possible that more digitally-driven teachers are overrepresented in this sample as less digitally-oriented teachers may have felt less comfortable answering the questionnaire. The convenience sample can explain the absence of the expected significant correlations.

Both in the planning of pedagogical action and intervention, physical, non-digital resources are the type of resource that teachers in the Azores continue to use most frequently, similarly to what was found by [1] in kindergarten and primary schools throughout Andalusia. Information that encourages the incorporation of technology into classrooms is easily obtained but rarely highlights the benefits of non-digital learning. Ref. [33] makes clear that non-digital teaching resources such as boards are the most popular and used visual aid as they are quick and easy to use and available in all teaching accommodation. Meaningful learning takes place through good communication practices between the learner and tutor [34] and digital resources cannot inhibit interaction between student and teacher. They cannot be seen as a substitute for a proficient teacher, but rather as beneficial as part of an integrative strategy to promote learning.

For planning, the curricular and programmatic guidelines in use, as well as textbooks, are preferred by teachers in the Azores. In pedagogical activities, textbooks appear again to be the most referenced resources, closely followed by activity sheets, an element of student work or assessment that just marginally reflects the learning construction processes. Many other pedagogical resources are also mentioned, far from the systematic and intentional use of digital resources.

The resources referred to by teachers are more directly linked to their own needs, and specific to tasks they perform as teachers, rather than taking account of students' needs, and their ways of learning. This therefore contradicts the current educational perspectives that give a more predominant role to learning and, consequently, to the student in this process [8,9].

Despite this still very traditional panorama and the apparent predominance of non-digital resources, many teachers state that digital platforms and web pages provide specific content for their practice. Their choice relies on relevant pedagogical information that can be quickly transposed into practice without major changes, avoiding those that they can build. This openness to digital, in particular the reference to platforms that present specific educational content, also reveals that teachers are beginning to recognize the relevance and pertinence of repositories that, by enabling a metadata-oriented search, can facilitate access to information and, perhaps, improve the quality of the learning processes they seek to enhance [2,3,21–24].

Primarily used for teaching content, some of the listed resources are mobilized to provide support materials to students or serve for students' research activities, but many are also used for the performance of assignments and students' assessments. These data seem to reveal a need for a paradigm shift in teachers, from mere exposition vehicles to pedagogical engagement. In fact, a focus on embedding pedagogical design into the open educational resource development process is a widely accepted trend [35].

Despite what is currently being assumed and the understanding of its importance by teachers, there still seems to be a tendency to be passive users rather than having a more participative role in LO creation for meeting the student needs. This practice, grounded in more theoretical classic didactical models, is, despite a general understanding of the role of practice-oriented activities in the learning process, in line with [19] findings that also stated that educators internationally may likewise use these new understandings as an empirical basis for their practice.

This idea is reinforced when we ask teachers to envisage the future use of teaching resources. Here, we reported a diversity of references to digital resources and conditions for their use that teachers would like to see come together for more consistent and widespread use of learning objects in their daily education. As many teachers in the region are available to use a digital repository in their classroom, thinking about its use seems to lead teachers to think in a logic of decentralization of learning processes, placing students also in a scenario of joint action and use, as pointed out in the studies by [3,25] or [28].

Finally, we conclude that this is still an area to be explored in educational terms. Although it has been a tool available to teachers and students for some decades, there still seems to be a lack of knowledge about its existence and potentialities. As [28] states, its use is closely linked to the development of digital skills and these skills should be part of teacher training, so this study also reinforces the need to invest in this area.

Moreover, [36] stressed the low availability of skills-based training programs concerning the pedagogical integration of ICT into teachers' practices. These programs rarely go beyond the traditional approach of showing teachers how to use or operate digital tools [37]. Workplace learning that includes experimental behavior may compensate for some of the problematic issues of training programs.

Experimental behavior is usually characterized by the inclusion of design-based components. According to [37], a design-based approach "affords teachers the opportunity to learn how to use the specific technologies in the context of their curricular needs". As a result, "teachers take more ownership of the resources [DLOs], have higher confidence in integrating them as teaching tools, and are more likely to believe that they will have a positive effect on student achievement". In short, it is suggested that experimental behavior with design-based components will enhance and optimize the use of digital LO. Also, embedding pedagogical design into the open educational resource development process is a widely accepted trend [38].

The assumed general use of non-digital and digital resources in Azorean schools is a starting point to promote a digital literate teachers' generation related to students. The use of digital tools can be a successful strategy to teach new generations who are very familiar with new technologies. However, despite the recognition of the learning benefits of digital LOs joint use by teachers and students, there still exists digital illiteracy, web connectivity and technical support issues to overcome to improve Los' integrated use and the students' learning outcomes.

In this sense, the results demonstrate the need to invest more in the teachers' initial and continuing training, enhancing the use of digital technology, by exploring in-depth the existing resources, their advantages, and how they can be implemented in pedagogical practices in favor of meaningful learning for students. In particular, it is important that this training looks at Learning Objects as a privileged gateway for the construction of knowledge by all those involved in the educational process, involving them in the development of literacy skills that are increasingly digital and technological.

As with any research, the study reported here presents some limitations. One of the main ones is related to the data collection instrument that, being a questionnaire to be filled out online, may have generated less commitment from the potential respondents in its completion. This is compounded by the fact that not all respondents are likely to have the same digital skills and interest to fill it out. Another limitation is the fact that it is not possible to generalize because it is a study carried out in a region with very specific characteristics, shaped by insularity and the ultra-peripheral nature of the location.

**Author Contributions:** Conceptualization, A.I.S., A.C.C., A.Z.B., M.I.P., J.C. and A.B.M.; methodology, A.I.S., A.C.C., A.Z.B., M.I.P., J.C. and A.B.M.; validation, A.I.S.; formal analysis, A.I.S., A.Z.B., A.C.C., A.B.M., A.R. and D.F.; investigation, A.I.S., A.C.C., A.B.M., A.Z.B. and M.I.P.; resources, A.B.M.; data curation, A.I.S., D.F. and A.B.M.; writing—original draft preparation, A.I.S.; writing—review and editing, A.C.C., A.Z.B., M.I.P., J.C., D.F., A.B. and A.B.M.; visualization, D.F. and A.I.S.; project administration, A.B.M.; funding acquisition, A.B.M. All authors have read and agreed to the published version of the manuscript.

**Funding:** This work is financed by the FEDER in 85% and by regional funds in 15%, through the Operational Program Azores 2020, within the scope of the project SEA-THINGS-Learning Objects for Promoting the Ocean Literacy (PO AÇORES 2020—ACORES-01-0145-FEDER00011) and had also the support of FCT through the strategic project UIDB/50027/2020 given to CIBIO/InBio that also involve FEDER funds through the Operational Programme for Competitiveness Factors. This work was also partially financially supported by Base Funding—UIDB/00027/2020 of the Artificial Intelligence and Computer Science Laboratory—LIACC—funded by national funds through the FCT/MCTES (PIDDAC).

**Institutional Review Board Statement:** The study was approved by the Ethics Committee of the University of the Azores.

**Informed Consent Statement:** Informed consent preceded the completion of the questionnaire and was presented in the following terms—"Dear Teacher, In the context of the project "SEA-THINGS—Learning Objects to Promote Ocean Literacy" (https://fgf.uac.pt/pt-pt/content/sea-things-objetos-de-aprendizagem-para-promover-literacia-oceanica; accessed between 20 September and 20 December 2021), developed by a group of researchers from the University of the Azores, we request your cooperation in completing a questionnaire that will allow us to survey the way the use of digital technologies in learning processes is seen, particularly in the promotion of ocean literacy by Educators and Teachers. Completing this questionnaire will take about 10 min, and you should answer the questions according to your professional reality and your perspective on the topics presented. Under the RGPD, it is clarified that the data collected in this form are for the exclusive use of this research work. The information collected respects the anonymity of the participants. If you do not agree with these conditions do not fill out the form. Any additional clarification or action you may require regarding your data, you may contact the person responsible: Armando B. Mendes, at armando.b.mendes@uac.pt.

**Data Availability Statement:** Not applicable.

**Conflicts of Interest:** The authors declare no conflict of interest.

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
