# Peer review of "Learning Objects in the Educational Context: The Perspective of Teachers in the Azores"

_education, doi:10.3390/educsci12050309_

Round 1
Reviewer 1 Report
The work under review is of interest to the educational field. However, there are some aspects that, in my opinion, the authors should clarify or revise:
1. it is not sufficiently clear to me what is the particularity (sociological or academic) that makes the analysis of the Azores case interesting. This should be made more explicitly clear in the introduction.
2. There is no clear definition of objectives or research questions, although the general purposes are intuited from reading the introduction. This should be added in the Materials and Methods section.
3. There is also no clear definition of variables, neither independent nor dependent. Within the participants, the authors distinguish certain characteristics that could act as independent variables (sex, area of knowledge, etc.), but it is not sufficiently clear to what extent they are authentic discriminating variables in the study. On the other hand, in the survey used there is a distinction of different parts. These parts could be understood as the dependent variables of the research, but this is not explicitly stated in the text either, as far as I have been able to see.
4. In relation to the survey, where does the classification of the questions into different blocks come from? Has a factor analysis been carried out? It would be useful to indicate the validation instruments that the authors used for their questionnaire.
5. The Materials and Methods and Results sections should be structured differently. Everything that has to do with the results of the study (including the results on independent variables that delimit the sociological and academic profile of the participants) should be included in the Results section. In the Materials and Methods section, the research objectives and a section to explain the research methodology used should be added.
Author Response
The authors would like to thank the reviewers for carefully reading the text and for all the suggestions and recommendations that made it possible to introduce relevant improvements and clarifications.
The changes made are as follows:
1. It is not sufficiently clear to me what is the particularity (sociological or academic) that makes the analysis of the Azores case interesting. This should be made more explicitly clear in the introduction.
Answer: we add a justification to clarify why the Azores Case study would be interesting in the paper context
2. There is no clear definition of objectives or research questions, although the general purposes are intuited from reading the introduction. This should be added to the Materials and Methods section.
Answer: hypothesis understudy was added to the Materials and Methods section.
3. There is also no clear definition of variables, neither independent nor dependent. Within the participants, the authors distinguish certain characteristics that could act as independent variables (sex, area of knowledge, etc.), but it is not sufficiently clear to what extent they are authentic discriminating variables in the study. On the other hand, in the survey used there is a distinction of different parts. These parts could be understood as the dependent variables of the research, but this is not explicitly stated in the text either, as far as I have been able to see.
Answer: Research hypothesis and identification of dependent and independent variables were added to the text.
4. In relation to the survey, where does the classification of the questions into different blocks come from? Has a factor analysis been carried out? It would be useful to indicate the validation instruments that the authors used for their questionnaire.
Answer: The classification of question blocks in the survey was aligned with the objectives, namely the type of resources used in class and knowledge about oceans, etc. Factor analysis of the results was performed but didn´t bring new interpretations, therefore the authors decided not to present any further treatment of the results, to avoid repetition of analysis. No validation instruments were used in the questionary design, as the presented results are part of a larger questionary and including validation, questions would make it too long and enhance the probability of its response outcome to perish.
5. The Materials and Methods and Results sections should be structured differently. Everything that has to do with the results of the study (including the results on independent variables that delimit the sociological and academic profile of the participants) should be included in the Results section. In the Materials and Methods section, the research objectives, and a section to explain the research methodology used should be added.
Answer: This has been done.

Reviewer 2 Report
The paper contributes to the actual and important question of contemporary education – how intensively do practicing teachers use digital learning resources, particularly learning objects. The particular case of the Azores islands provides an additional interest to the paper. The main strength of the paper is a profound analysis of various types of resources that teachers use together with the reasons for their choice in the planning of pedagogical work.
The paper has a clear structure. All illustrative material is appropriate.
There are several specific comments on the paper:
- References section contains about 1/3 of sources within the last 5 years (since 2018). Therefore, this might be not sufficient according to the Journal recommendations.
- Author (s) formulate research objectives in the Introduction. However, adding a clear a hypothesis would be also beneficial for the reader to understand the methodology and research results. Consequently, in the end of the paper the author (s) could return to the hypothesis and conclude weather it has been proved or not.
- Several question samples or a more detailed questionnaire template should be added to the main text or to an annex of the article. That would help the reader to see exactly how the “open- and closed-ended questions” questions were formulated and would make the manuscript’s results reproducible.
- A more clear discussion of the limitations and perspectives of the study is recommended.
- The author (s) give clear conclusions that show “how pedagogical resources are used by teachers in the Autonomous Region of the Azores”. However, it would be very interesting if they also could add several recommendations to the teacher training institutions – what exactly should be reinforced in training or internship programmes to bridge the still existing second digital divide; or give some examples what has already been done in this context in the Azores Islands.
Author Response
The authors would like to thank the reviewers for carefully reading the text and for all the suggestions and recommendations that made it possible to introduce relevant improvements and clarifications.
The changes made are as follows:
1. References section contains about 1/3 of sources within the last 5 years (since 2018). Therefore, this might be not sufficient according to the Journal recommendations.
Answer: Literature sources were updated as suggested, with the inclusion of several very recent publications.
2. Adding a clear a hypothesis would be also beneficial for the reader to understand the methodology and research results.
Answer: Hypotheses were added.
3. Consequently, in the end of the paper the author (s) could return to the hypothesis and conclude weather it has been proved or not.
Answer: considerations on the hypothesis were added at the beginning of conclusions’ section.
4. Several question samples or a more detailed questionnaire template should be added to the main text or to an annex of the article.
Answer: Some more information about the questionnaire was added to the text. The questionnaire is available for consultation upon request to the first author.
5. A more clear discussion of the limitations and perspectives of the study is recommended.
Answer: A clearer and more detailed analysis the limitations and perspectives of the study was included in the conclusions as suggested.
6. It would be very interesting if they also could add several recommendations to the teacher training institutions – what exactly should be reinforced in training or internship programs to bridge the still existing second digital divide; or give some examples what has already been done in this context in the Azores Islands.
Answer: Recommendations were included in the conclusions section as suggested.

Round 2
Reviewer 1 Report
In my opinion, the authors have responded adequately to the questions and comments. As a consequence, the quality of the manuscript has increased. I have only a few suggestions:
- I think that the general objective that the authors have introduced in section 3.1 should be broken down into several specific objectives. In these specific objectives, the intention of the study to differentiate by academic level (early childhood, primary or secondary education) and area of knowledge should be pointed out.
- Section 3.2 should be titled "Instruments and procedure".
- It would be interesting to add in the Discussion section a paragraph discussing the results obtained with other similar studies carried out at the university level. I suggest that the authors incorporate, in this respect, references such as the following:
https://doi.org/10.3390/app112411649
https://doi.org/10.30827/publicaciones.v52i3.22270
https://doi.org/10.3389/fpsyg.2021.617650
Author Response
Regarding the reviewer's comments, we are pleased to acknowledge that he considers that the paper was improved. Concerning the new suggestions:
1. I think that the general objective that the authors have introduced in section 3.1 should be broken down into several specific objectives. In these specific objectives, the intention of the study to differentiate by academic level (early childhood, primary or secondary education) and area of knowledge should be pointed out.
Answer: Specific objectives are detailed as suggested.
2. Section 3.2 should be titled "Instruments and procedure".
Answer: Section 3.2 was renamed as suggested.
3. It would be interesting to add in the Discussion section a paragraph discussing the results obtained with other similar studies carried out at the university level. I suggest that the authors incorporate, in this respect, references such as the following:
https://doi.org/10.3390/app112411649
https://doi.org/10.30827/publicaciones.v52i3.22270
https://doi.org/10.3389/fpsyg.2021.617650
Answer: A paragraph was added in the discussion section discussing the results obtained with other similar studies carried out and one of the suggested references was incorporated. The other two were considered difficult to relate to our results and were disregarded. However, another recent study was referenced.